



**Extension and Inversion of Salt-Bearing Rift Systems**
Tim P. Dooley & Michael R. Hudec
*Applied Geodynamics Laboratory, Bureau of Economic Geology, Jackson School of*
*Geosciences, University Station Box X, Austin Texas 78713, USA*
**Abstract:**
*We used physical models to investigate the structural evolution of segmented extensional rifts*
*containing syn-rift evaporites and their subsequent inversion. An early stage of extension*
*generated structural topography consisting of a series of en-échelon graben. Our salt analog*
*filled these graben and the surroundings before continued extension and, finally, inversion.*
*During post-salt extension, deformation in the subsalt section remained focused on the graben-*
*bounding fault systems whereas deformation in suprasalt sediments was mostly detached,*
*forming a sigmoidal extensional minibasin system across the original segmented graben array.*
*Little brittle deformation was observed in the post-salt section. Sedimentary loading from the*
*minibasins drove salt up onto the footwalls of the subsalt faults, forming diapirs and salt-ridge*
*networks on the intra-rift high blocks. Salt remobilization and expulsion from beneath the*
*extensional minibasins was enhanced along and up the major relay/transfer zones that separated*
*the original sub-salt grabens, forming major diapirs in these locations.*





*Inversion of this salt-bearing rift system produced strongly decoupled shortening belts in*
*basement and suprasalt sequences. Suprasalt deformation geometries and orientations are*
*strongly controlled by the salt diapir and ridge network produced during extension and*
*subsequent downbuilding. Thrusts are typically localized at minibasin margins where the*
*overburden was thinnest and salt had risen diapirically on the horst blocks. In the subsalt*
*section, shortening strongly inverted sub-salt grabens, which uplifted the suprasalt minibasins.*
*New popup structures also formed in the subsalt section. Primary welds formed as suprasalt*
*minibasins touched down onto inverted graben. Model geometries compare favorably to natural*
*examples such as those in the Moroccan High Atlas.*
**1.      Introduction**
As noted by Bonini et al. (2011), in their review paper, "basin inversion" is a commonly used
term to signify shortening of formerly extensional basins (cf. Buchanan and McClay, 1991;
Buchanan and Buchanan, 1995;  Ziegler, 1987). Localization of shortening by extensional rifts,
and their subsequent inversion, is not surprising as these are long-lived crustal weak zones.
Inversion of graben and entire rift systems has been a significant focus of study since the early
1980s owing to its importance related to: (1) the role of pre-existing faults in focusing and
accommodating shortening of the upper, shallow crust; (2) the role of pre-inversion high-angle
faults as potential seismogenic sources and hazards, and; (3) their economic importance related
to focused fluid flow and associated ore deposit generation was well as influencing hydrocarbon
maturation, migration pathways, and trapping in inverted petroleum-bearing sedimentary basins
(see Bonini et al., 2011 for further details and references). Deposition of evaporites in these




systems, either as syn-rift deposits or immediately after rifting, can add complexity to the system
in many ways. For example salt may have significant variation in thickness across the rift
resulting in varying degrees of coupling between the basement and suprasalt sediments during
subsequent extension and inversion (e.g. Withjack and Calloway, 2000). Salt may also be
expelled from beneath depotroughs, during extension and/or loading to form diapir networks that
may later focus shortening as plate motions evolve (e.g. Dooley et al., 2005). These diapir
networks may be surrounded by patchy weld systems adding further complications to the system
(cf. Rowan and Krzywiec, 2014).

Examples of basement-involved inverted salt-bearing rifts include the the Mid-Polish Trough
(e.g. Krzywiec, 2012; Rowan and Krzywiec, 2014), the southern North Sea (e.g. Stewart, 2007;
Stewart and Coward, 1995; Davison et al., 2000; Jackson and Stewart, 2017) and the High Atlas
of Morocco (e.g. Saura et al., 2014; Martín-Martín et al., 2016; Moragas et al., 2017; Teixell et
al., 2017; Verges et al., 2017; Figure 1). The central High Atlas range is a doubly-vergent fold-
thrust belt that formed by inversion of a Triassic-Jurassic rift basin during the Alpine orogeny
(e.g. Teixell et al., 2003; Saura et al., 2014; Moragas et al., 2017). Within the central part of the
range outcrop is dominated by Lower-Middle Jurassic deposits that form brand synclines or flat-
topped plateaux, and separated by NE-SW oriented anticlines or thrust faults (Figure 1; Moragas
et al., 2017). These ridges have had a variety of explanations for their origin such as
transpressional deformation or the emplacement of Jurassic intrusions (see detailed discussion in
Moragas et al., 2017, for more details). A few studies of individual structures proposed a diapiric
origin for these ridges (e.g. Michard et al., 2011). However, more recent studies have interpreted
the entire Central High Atlas as a complex salt-bearing rift basin with associated diapirism and



minibasin formation, that was inverted in the Alpine orogeny. For example, Saura et al. (2014)
documented more than ten elongated extensional minibasins that were originally separated by,
now welded, salt walls. Thick evaporitic successions were deposited within the developing rift in
the late Triassic (Verges et al., 2017). Extension continued into the Early Jurassic with coeval
diapirism and minibasin formation, followed by a long post-rift stage where halo kinetic
processes continued to evolve (Moragas et al., 2017; Martín-Martín et al., 2017). Inversion began
in the late Cretaceous (e.g. Verges et al., 2017), squeezing a complex diapir and minibasin
province. Such a diapir and minibasin province is likely to exhibit extreme variations in
overburden strength, and thus behavior, during shortening. It is this salt-tectonic scenario that
formed the inspiration for our experimental study.

Some previous physical modeling studies of basement-involved extension and inversion of salt-
bearing rifts include those of Dooley et al. (2005) with application to the North Sea, and Moragas
et al. (2016) in their focused study on syn- and post-rift diapirism and inversion in the Moroccan
High Atlas. Bonini et al. (2011) modeled detached extension and subsequent shortening of these
graben, and Roma et al. (2017, 2019) as well as Ferrer et al. (2016) modeled extension and
inversion above rigid planar and ramp-flat extensional master faults with high-level salt layers.
However, all the basement-involved studies to date relied on non-deformable basement blocks to
generate extension and subsequent inversion. An exception to this are the clay models of
Durcanin (2009), but these models could not be sectioned and thus sections shown in this study
are "hypothetical". A new series of experiments was designed to produce segmented rift systems
in deformable model materials, fill them with syn-rift evaporites and subject them to further
extension, loading and, finally, inversion. Our goals with these models was to test: (1) where and



why do diapirs form in a segmented extensional rift system?; (2) how much coupling is there
between basement and cover separated by a relatively thick salt body during extension and
contraction?; (3) what styles of shortening structures form in the suprasalt section during
inversion and what controls their location and style?, and; (4) what are the styles of shortening in
the subsalt section and can we get significant reactivation of extensional structures during
inversion?

**2.    Modeling Methodology**

*2.1 Model Design and Scaling*

Our goal with these models was to generate a series of en-échelon graben across a rift system in
a similar fashion to models presented in Dooley et al. (2005). They achieved this by using non-
deformable wooden blocks with a series of steps, whereas we wished to generate segmented rifts
using deformable materials that could be serially sectioned at the end of the model run. Previous
models of segmented rifts systems used offset rubber sheets to do this (e.g. McClay et al., 2002;
Amilibia et al., 2005). However, these suffered from internal artifacts as the rubber is stretched
to generate extension in the overburden it also constricts orthogonal to the extension direction,
resulting in accommodation or transfer zones that are structural lows rather than highs in these
locations (e.g. see Sections 2 and 5 of Figure 8 in Amilibia et al., 2005). In order to mitigate
these effects we used a hybrid system comprising a single basal stretching sheet, a thin basal
silicone detachment and a series of polymer slabs to generate a segmented rift system in the
overburden (Figure 2). The stretching rubber sheet generated extension, whilst the basal polymer





layer acted as an efficient detachment (during extension and contraction). In contrast, the
polymer slabs served to focus extension at these sites, in much the same way that precursor
diapirs focus strain in contractional models (e.g. Dooley et al., 2009, 2015; Callot et al., 2007,
2012). Dooley and Schreurs (2012) employed a variety of polymer "crustal weak zones" to focus
extension in pull-apart basins and to concentrate and perturb deformation above basement strike-
slip zones. Le Calvez and Vendeville (2002), Zwaan et al. (2016) and Zwaan and Schreurs
(2017) also used polymer "ridges" to focus or "seed" extensional structures in their models, and
Marques et al. (2007) used wedge shaped polymer layers to investigate transform faulting
associated with ridge push. Dual motors generated the symmetric extension and contraction in
these models (Figure 2).

Models are dynamically scaled such that 1 cm in the model approximates to 1 km in nature (see,
for example, Brun et al. 1994 and McClay 1990 for detailed discussions on scaling). Models
were conducted with combined horizontal velocities of $1.4 \times 10^{-4}$ cm/s that yields a strain rate of
$1.8 \times 10^{-6}$ s$^{-1}$. This rate models an extensional fault system with a moderate displacement rate
(e.g. Withjack & Callaway 2000; Dooley et al., 2005). More importantly, post-salt extension was
pulsed in order to allow the model salt analog to react to the imposed strain and the differential
loads induced by spatially-variable thickness of the synkinematic sediments added after each
increment of extension. Models consist of 3 or 4 main evolutionary stages (Table 1): (1) pre-salt
extension followed by addition of model salt into the main structural topography addition of a
regional salt fringe and thin roof: (2) post-salt extension delivered in a series of pulses, as
described above; (3) post-salt loading and downbuilding stage, allowing diapirs that formed in
Stage 2 to continue to rise vertically, and; (4) inversion, where the moving endways are detached



from the baseplates, the baseplates are clamped in place, and motion is reversed. We focus
primarily on the results of one experiment (Model 1, Table 1) in the descriptive sections and use
some of the results from two other experiments (Models 2 and 3, Table 1) to discuss salt
tectonics styles and salt migration pathways in non-inverted and weakly inverted rifts in the
discussion section. The only difference between Model 1 and Models 2 and 3 is that the thin
basal detachment layer extended across the entire model base in Model 1, whereas it was limited
to just covering the rubber sheet in Models 2 and 3 (see Figure 2).

*2.2 Modeling Materials*

As with other physical modeling studies of salt tectonics, we simulated rock salt using ductile
silicone and its siliciclastic overburden using brittle, dry, granular material. The silicone was a
near-Newtonian viscous polydimethylsiloxane. This polymer has a density of 950 to 980 kg m$^{-3}$
and a dynamic shear viscosity of $2.5 \times 10^4$ Pa s at a strain rate of $3 \times 10^{-1}$ s$^{-1}$ (Weijermars, 1986;
Weijermars et al., 1993). In some of our models the salt analog was dyed with minute quantities
of powdered pigments in order to track salt flow paths in the completed model. The layered
brittle overburden comprised different colored mixtures of silica sand (bulk density of ~1,700 kg
m$^{-3}$; grain size of 300-600 μm; internal friction coefficient, μ, = 0.55–0.65; McClay, 1990;
Krantz, 1991; Schellart, 2000), and hollow ceramic microspheres ("glass beads") having a bulk
density of 650 kg m$^{-3}$, average grain size 90-150 μm, and typical μ = 0.45 (e.g. Rossi and Storti,
2003; Dooley et al., 2009).



The hollows spheres serve to lower bulk grain size, as well as allowing us to modify the density
of the brittle overburden. Most physical models of salt tectonics have a layered brittle
overburden of pure quartz sand, which creates density ratios that are much higher than those of
nature. Exaggerated density ratios erroneously magnify overburden foundering, rise of active
diapirs, and expulsion and extrusion of salt (Dooley et al., 2007, 2009). In our models, the pre-
rift overburden sediments had a density ratio of equal to that of our model salt by our varying the
sand-bead ratio in the brittle section. This was done to minimize any density- or buoyancy-driven
rise of the basal slabs that are also made of the same materials as our salt analog. In stages 2 and
3 of the model runtime the density of the sedimentary load was increased to 1.1-1.2 times that of
our model salt. This was done to encourage salt remobilization from beneath the extensional
minibasins in Stage 2 and to keep salt structures (diapirs) growing in Stage 3.

*2.3. Data Capture, Visualization, and Interrogation*

Computer-controlled cameras photographed the obliquely lit upper surface of the models at set
time intervals. A digital image correlation (DIC) system, consisting of a high-resolution stereo
charge-coupled device (CCD) system and associated software, tracked the surface-strain history,
subsidence, and uplift values, as well as displacement vectors of the top surface of the model.
Adding synkinematic layers means data is incremental for these data. For more details on DIC
monitoring techniques, see Adam et al. (2005). After completion models were impregnated with
a gelatin mixture, left to partially dry for 12 hours and then sliced into closely spaced slabs.
Coregistered digital photographs of these closely spaced serial sections (≤3.5 mm apart) yielded
a 3D voxel model of completed model. Dip sections are the sliced and photographed cross





sections, whereas crosslines, arbitrary lines and depth slices are virtual sections constructed from
the voxel model. As a result, the crossline, arbitrary line and depth slice images are interpolated
and thus not as sharp as those derived directly from photographed dip sections. In addition the
3D salt volume can be extracted from this voxel by coloring the salt in each section with a
known pixel value (e.g. white for a value of 255).

**3.        Experimental Results**

*3.1 Stage 1: Pre-Salt Extension*

Stage 1 comprised 3 cm of uniform extension in order to generate structural topography that was
infilled by our salt analog (Table 1). The basal weak slab array shown in Figure 2, was there to
ensure a segmented rift system formed. Height-change data ($\Delta Z$; Figure 3a) generated from our
stereo-DIC system reveals the main rift system in Model 1 comprising en-échelon graben that
step to the right across the underlying basal slab array (Figure 2). Three main depotroughs are
seen along the segmented rift system, separated by zones of higher intra-rift topography,
accommodation zones (Figure 3). Strain data illustrate the focused extension along the fault
network across the rift system (Figure 3b). On many faults maximum extensional strains, and
maximum width of faults, are recorded along their centers, although some deviate from this trend
(Figure 3b).  Weaker extensional systems form at the margins of the model, far from the central
rift system (Figure 3). The accommodation zones are clearly seen in the strain data, and
consisted of interlocking arrays of mostly soft linked extension faults with some rotation seen at
fault tips (Figure 3b). Between the southern and central and between the central and northern



subbasins, clear fault-tip rotation is seen with breaching of the major relay systems separating the
subbasins (Figure 3b).

After this stage, our salt analog was placed into the three subbasins and allowed to settle (Figure
4a). Once this had settled and filled the structural relief a 12-mm-thick regional layer of our salt
analog was emplaced across the model as a series of tiles (Figure 4b), and allowed to degas prior
to Stage 2.

*3.2 Stage 2: Post-Salt Extension*

Our salt analog in Model 1 was buried under a thin (4 mm) sedimentary roof before undergoing a
further 6 cm of extension during Stage 2 (Table 1). Figure 5 shows height-change data and strain
from Model 1 after applying total of 4 cm of post-salt extension. Synkinematic sediments were
added after each 1 cm of basement extension and the values shown in Figure 6 are incremental
for that phase of extension, i.e. 3-4 cm post-salt extension. During this period the main
depotrough comprised a sigmoidal extensional minibasin located above the original offset graben
system (Figure 5a). A series of curvilinear fabrics define relatively minor surface faulting (Figure
5a). Strains seen on the upper surface were much more diffuse and spread across the rift system
than those seen in the pre-salt extension stage (Figure 3). The strain fields formed curvilinear
systems of extension that, for the most part, defined minor graben above reactive diapirs, and
appear to be diagnostic of detached suprasalt extension (cf. Dooley et al., 2005; Figure 5b).
Maximum extensional strains were seen adjacent to the sigmoidal depocenter, as expected,
delineating the margins of the main depotrough, and in locations that were accommodation zones



during the pre-salt extension phase as the cover collapsed into the developing trough (Figures 3
and 5). Minor shortening strains are seen within the extensional minibasin/depotrough due to
inner-arc contraction as it subsided into, and expelled, the salt (Figure 5b). The marginal graben
at the ends of the model continued to subside during this stage (Figure 5).

*3.3 Stage 3: Post-Extension Loading*

Model 1 underwent 9 cm total extension prior to moving on to a downbuilding or post-extension
loading phase in Stage 3. Stage 3 lasted for 5 days and synkinematic sediments were added daily,
keeping apace and gently covering any positive topography that developed whilst continuing to
load negative topography.

Height change maps of the model surface of Layers 1 and 4 are shown in Figure 6. Clearly
illustrated in Figure 6a are the rising diapir networks as salt was expelled from beneath the
composite minibasin in the model center.  Comparing Figure 6a to the strain map in Figure 5b
one can immediately see that the diapir networks closely conform to the strain patterns seen
during Stage 2, evolving from reactive to passive features in this post-extension stage. Diapirs
labelled 1-3 are all located on the footwalls of the main extensional minibasin, and, more
importantly, in locations that lie above, and along, what were the original accommodation zones
between the original subbasins (see Figures 3, 5 and 6). More linear salt walls are seen rising
adjacent to the marginal graben systems and the extensional minibasins is flanked by upwellings
along most of its length (Figure 3). Figure 6b illustrates the height change map after 4 days into
Stage 3. Activity waned in these systems over time except for the more active and emergent





diapirs (1 & 2 in Figure 6b). Smaller amounts of salt rise are seen flanking the central region of
subsidence.

*3.4 Stage 4: Inversion*

In Stage 4 Model 1 was covered with a thin roof sequence and subjected to 25 cm of lateral
shortening (Table 1). Height-change maps reveal the evolution of the model during inversion
(Figure 7). As expected from previous studies (e.g. Dooley et al., 2009, 2015; Callot et al., 2007,
2012; Duffy et al., 2018), initial shortening resulted in rejuvenation of the two main diapirs
formed during the extension and loading stages (1 and 2 on Figure 7a). This was followed by
uplift of the composite minibasin system and the formation of a series of linear and curvilinear
uplifts (Figure 7b). These uplift patterns are very similar to the ridge networks seen during Stage
3 (compare Figures 6a with 7b). With continued shortening the minibasin system continued to
rise and salt emerged from Diapir 2 (Figure 7c). The network of curvilinear flanking uplifts
continued to rise and become more prominent, and intervening lows shrank in area as they were
overthrust (Figure 7c). At the end of the experiment Model 1 consisted of a central plateau that
was cored by the minibasin system, and flanked by linear and curvilinear thrust ridges with
narrow intervening lows (Figure 7d). Salt sheets emerged from Diapirs 1 and 2 and flowed down
into the flanking topographic lows (Figure 7d).

The final overhead view of Model 1 is shown in Figure 8. In this we see the central uplifted
minibasin system forming an oblique plateau across the model, and flanked by the linear and
curvilinear faulted ridge network. Flow directions of the salt sheets emanating from Diapirs 1





and 2 are indicated by red arrows (Figure 8). Major fault scarps were partially degraded exposing
older strata, and some scarps abut, or override scarps with opposite sense of dip (Figure 8). On
the right side of the model two fault scarps abut in the south and then coalesce forming a very
narrow fault zone (Figure 8). These geometries and relationships are revealed by a series of four
sections through Model 1 (Figure 9). The four sections illustrate the decoupled nature of
deformation between sub- and suprasalt strata (Figure 9). The main feature is the structurally
elevated extensional minibasin system that trended obliquely across the model (Figures 8 and 9).
For much of the strike length this feature is flatted topped, and bounded on either side by
detached suprasalt thrusts or secondary thrust welds as Diapirs 1 and 2 were squeezed shut
(Figures 8 and sections 33 and 55 in Figure 9). Structural elevation of this minibasin system was
partly aided by the inversion of subsalt graben that form inversion anticlines and harpoon
structures in the subsalt strata (Figure 9; see the next section for further discussion). Primary
welds denote where the minibasins have touched down on the subsalt strata. Also of interest in
the suprasalt strata are emergent sheets and isolated salt bodies sourced from the squeezed and
welded diapirs (e.g. section 33 and 55 in Figure 9), salt-cored thrusts and related secondary
welding of portions of these as salt was ejected and hangingwall touched down onto footwall
(e.g. Sections 33, 86 and 106, Figure 9). Other structures in the suprasalt section include highly
overthrust popdowns, and narrow upright fault zones as hangingwalls collided during shortening
(Section 86, Figure 9). A curious structured is observed in many sections, termed an "S"
structure due to its shape (see Sections 55 and 86 in Figure 9). We will discuss the origins of this
structure in the discussion section. In the subsalt strata structures are very different, consisting of
inverted and heavily deformed graben systems in both the center and margins of the model as
well as new popup structures (Figure 9).






## 4.    Discussion


In this section we focus on: the formation and location of diapirs during extension and post-
extension loading; shortening styles and location in the suprasalt section during inversion;
shortening styles and locations in the subsalt section, and; comparison of model results to
examples from the High Atlas in Morocco.

*4.1 Diapir Formation and Location During Extension*


In Model 1 the main diapirs (diapirs 1 and 2, Figure 6), and associated salt wall or ridge
networks formed in the footwall of the main extensional systems that flanked the composite
extensional minibasin. More specifically the most active diapirs formed in locations spatially
associated with the interlocking accommodation zones that originally separated the subbasins
(Figures 3, 5 and 6). These locations are similar to those documented in Dooley et al. (2005),
although the transfer zones in those models were vertical and rigid. Model 2 was run with almost
identical parameters as Model 1, but was not inverted, preserving the diapir geometries and
locations (Table 1 and Figure 10). Figure 10a shows the height-change map that evolved during
Stage 1 of this model (see Table 1), consisting of an en-échelon series of three graben that run
obliquely across the model, similar to that seen in Model 1 (see Figure 3a). The only difference
Model 2 showed was the presence of marginal graben that formed closer to the main rift system
than that seen in Model 1. This was attributed to the narrower basal silicone detachment used in
Model 2 (Figure 2 and Table 1). Likewise the continued evolution of Model 2 through Stages 2



and 3 was very similar to that seen in Model 1 (compare Figures 10b, c with Figures 5a and 6a).
The diapir network geometries and most active diapirs in Model 2 were very similar to those
seen in Model 1.

A section from Model 2 illustrates the extensional minibasins formed above the main graben and
diapirs located in the footwalls of these graben (Figure 10d). As we saw in the strain maps for
Model 1 there was only minor discrete extension in the suprasalt strata and these are cored by
reactive diapirs (Figure 10d). The main diapir in this section is located in the footwall of the
main graben system, and just along strike from the accommodation zone that separated the
southern and central subbasins (Figure 10a, d). Salt expelled from beneath the subsiding
minibasin flowed up onto the footwall and helped feed this growing salt diapir. We believe that
salt was also preferentially expelled up and along the accommodation zones that separated the
original subbasins and into these growing diapirs, as these accommodation zones have more
gentle relief compared to the steep faults that bounded the minibasins, thus offering a more
efficient conduit for salt flow.

In order to corroborate this concept of preferred flow up and along transfer or accommodation
zones, images from a third model, Model 3, are shown in Figure 11. Model 3 was subjected to
the same amount of extension as Model 1, but a very limited amount of inversion (Table 1). In
addition, the lack of a basal detachment across the entirety of the model base meant that
shortening in the subsalt section was limited to shortcut thrusts close to the margins of the
deformation rig that transferred shortening up to the weaker suprasalt section, with minimal
shortening seen in subsalt strata in the central portion of the rift system (Figure 11a-e; Table 1).



The lack of deformation in the subsalt strata means that primary welds seen in the sections in
Figure 11c-e, occurred during extension rather than during shortening. Depth slices from Model
3 illustrate the composite, stepped, minibasin that formed above the en-échelon rift system
(Figure 11a-b). The yellow marker salt that initially occupied the central graben of the rift
(Figure 11f) is seen to be expelled up and out of this graben system into the footwall, where it
helped inflate reactive diapirs that initially formed along these locations (Figure 11a-c; see
reactive diapir on the right side Figure 10d for a non-inverted example). Model 3 also had
substantial diapirs that flanked the rift in similar positions to those of Models 1 and 2, and yellow
marker salt is seen to flow along and up the, now faulted, lower-relief accommodation or transfer
zones and into these diapirs (Figure 11d-e). Thus, salt flow during extension and post-
extensional loading in Model 3 was multidirectional, being driven by differential loading out and
up onto the intra-rift horst blocks both up the main subsalt faults and along lower-relief pathways
such as the transfer zones that separated the subbasins in this rift system. Flow up and along
these conduits was eventually curtailed or stopped by primary welding (Figure 11c-e).

*4.2 Shortening in the Suprasalt Section*

Figure 12 shows the salt volume that was extracted from the serial sections and exported as a
point cloud. This image beautifully illustrates the structural style in the shallow section. The
central part of the model is dominated by the inverted extensional minibasin system that forms
an oblique structural low. Primary welds are denoted by gaps in the data, as the subsalt graben
were inverted and structurally elevated the minibasin system (Figure 12). The minibasin is
flanked by outward-vergent salt-cored thrusts, thrust welds and remnant high-level salt bodies or



sheets (diapirs 1 and 2, Figure 12). Thrust vergence reverses toward the margins of the model,
and structures vary from salt-cored thrusts to box-like thrusted folds (Figure 12).

Shortening in the suprasalt section is primarily controlled by the diapir and ridge network that
formed during extension and post-extensional loading. This is clearly illustrated in Figure 13,
which shows a height-change map, depth slice and dip section from Model 1. The diapir-ridge
networks, labelled a-e on Figure 13, localized shortening structures because these are where the
overburden was thinnest and thus weakest, and the diapir networks helped to focus deformation.
Deformation in the shallow section is clearly detached from the subsalt structures, except where
the minibasin system welded down onto inverted subsalt graben (Figures 9, 12 and 13c) Height-
change maps from the inversion phase also illustrate this reactivation of the pre-inversion diapir-
ridge network (compare Figure 7 and 13). Minibasin subsidence patterns in Model 1 were
primarily symmetric during extension and post-extensional loading stages as evidenced by
height-change maps (Figures 5 and 6), and by the stratal geometries seen in cross sections
(Figure 9). During inversion the main minibasin system was structurally uplifted by the inverting
subsalt graben (Figures 7), with little or no internal deformation except at the minibasin margins
where suprasalt thrusts developed (Figures 9 & 13c). Only minor tilting caused by shortening of
the main minibasin system is seen in the northern part of the model (section 106, Figure 9).
Smaller minibasins developed above the marginal graben systems exhibit more severe tilting as
they were carried up in the hangingwalls of major suprasalt thrusts (sections 33 and 55, Figure

388    9).


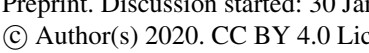


390 As mentioned in Section 3.4, there is a curious structure observed in suprasalt strata in some

391 sections through Model 1 that is termed an "S" structure due to its shape (sections 55 and 86,

392 Figure 9). This structure is found in a deformed salt-cored box fold in the southern part of the

393 model (Figures 9 and 12). A series of sections through this structure give a pseudo-temporal

394 evolution of this structure (Figure 14). The structure started as a faulted box fold, localized along

395 a salt wall (e in Figure 13), that initially formed during extension and post-extension loading.

396 One of the hinges began to fail on one side of this box fold and eventually limb failure occurred,

397 forming a small weld as the core began to narrow (Figure 14a-b). Eventually salt in the core was

398 expelled and the limbs welded, leading to the "S" geometry (Figure 14c).

399

400 *4.3 Shortening in the Subsalt Section*

401

402 We noted briefly in Section 3.4 that deformation in the subsalt strata is very distinct from that

403 seen in suprasalt strata (Figure 9). In subsalt strata the most obvious structures are the popup

404 structures in cross-section views, and none of these are linked to structures in the shallow section

405 (Figure 9). However the most interesting structures are found along the central portions of the

406 model where the pre-salt graben have been strongly deformed and inverted (Figure 9). Some of

407 these structures form the highest subsalt relief seen in Model 1 (e.g. section 33, Figure 9).

408 Height-change maps from Stage 4 of Model 1 clearly illustrate that the main minibasin system

409 was preferentially uplifted as an intact block during shortening (Figure 7). This uplift was thus a

410 result of preferential inversion of the subsalt graben system. This is borne out in cross-section

411 views through Model 1 that clearly show uplift of the main minibasin system as a coherent block

412 forming an almost flat plateau along the length of the center of Model 1 (Figures 7, 8 and 9).



Figure 15 shows detailed views of a non-inverted graben from Model 2 and an inverted graben
from Model 1. In Model 2 the non-inverted section the structure consists of a mildly asymmetric
graben with a smaller keystone graben formed against the more dominant right-side boundary
fault (Figure 15a). Figure 15b shows a portion of Section 33 from Model 1 (Figure 9). In it we
see a highly inverted basement graben system with a keystone graben system on the right margin
as in Model 1 (Figure 15a-b). Inversion of this graben was asymmetric with greater uplift of the
left side forming a harpoon-like inversion anticline that structurally elevated the suprasalt
minibasin (Figure 15b). Based on the geometry of the non-inverted model the right side of the
graben also saw significant inversion before being overthrust by a new subsalt thrust (Figure
15b). More minor new thrusts are seen to the left of the graben system. Only a minor amount of
structurally-induced tilting is seen in the suprasalt sequence (2.5°, Figure 15b), attributed to the
primary weld being slightly off the mid-point of the minibasin. Detached outward-vergent thrusts
are located at the minibasin margins (Figure 15b).

As noted by Amilibia et al. (2005), amongst others, inversion of normal faults in laboratory
models using sand is quite limited, sometimes being seen at shallow fault tips but bypass or
shortcut faults are far more common. Fault reactivation in nature can occur under stress levels
lower than that required to initiate new faults (e.g. Sibson, 1995), due to preexisting faults having
a lower cohesive strength and friction coefficient than that of intact rock (Anderson, 1951). The
lack of significant reactivation in sandbox models can be explained by the relative lack of
difference between the strengths of faulted and unfaulted sands, favoring the formation of new
shortcut faults (see Amilibia et al., 2005, and Bonini et al., 2011, for more details). Significant
reactivation of graben-bounding faults in our models (see Sections 33 and 106 in Figure 9;





Figure 15b) are attributed to two factors. The first is the presence of the weak basal slabs that
initially focused extension. Figure 15a shows remnant 'horns' of the polymer on either side of
the graben, and during shortening these would help to focus initial shortening onto the graben
system in the central part of the model. The second, and likely more important, reason is
interstitial infiltration of polymer into a narrow zone of the brittle section forming a hybrid
rheology along the preexisting faults. This results in a slight change in color of the granular
materials at the sand-silicone interface, which is just visible in Figure 15. Prior to inversion the
base and sides of the graben were in contact with silicone, resulting in interstitial infiltration
(Figure 15a). The upper portions of the graben-bounding faults were also in contact with silicone
again allowing for interstitial infiltration (Figure 15a). During shortening this interstitial
infiltration acted as a "lubricant" allowing reactivation and inversion of these faults (Figure 15b).

*4.4 Comparison to Examples from the Moroccan High Atlas*

Saura et al. (2014) documented that inversion-related deformation in the central High Atlas of
Morocco is mainly focused on minibasin margins with little internal deformation of these
minibasins, with diapirs that originally separated these extensional minibasins soaking up much
of the deformation, as is seen in our Model 1 (Figure 9). One such example is the Amezraï
minibasin (Figure 16a; Saura et al., 2014; Moragas et al., 2017; see location on Figure 1). As in
our Model 1, the Amezraï minibasin formed above a basement graben system and was flanked
by complex diapirs located in the footwall of this graben system (Figure 16a-b). After inversion
these flanks are the sites of significant upturn of flanking strata, thrusts welds and remnant
pedestals, similar to structures found in Model 1 (Figure 16a-b). The Azag minibasin lies further





to the ENE along the central High Atlas (Figures 1 and 16c; Teixell et al., 2017). Again, this
minibasin formed above a basement graben or half-graben system before being it was caught up
in Alpine shortening resulting in the welding of adjacent diapirs (thrust or secondary welds;
Figure 16c; Teixell et al., 2017), as seen in Model 1. The Azag minibasin also displays
significant tilting in the E-W cross section of Figure 16c. Minibasins can tilt during subsidence
either before or after welding (Rowan and Weimer, 1998; see Jackson et al., 2019, for more
details), but the stratigraphic architecture of the Azag minibasin consists primarily of bowl- and
tabular-shaped units indicating relatively symmetric subsidence during minibasin growth.
Significant tilting of minibasins caused by shortening is seen in some locations in our Model 1
(e.g. Section 106 of Figure 9), and thus, by analogy, the tilting and welding seen in the Azag
minibasin is attributed to Alpine shortening and basement uplift.

One notable difference between our model results (Figures 9 and 16b) and the example sections
shown in Figure 16a, c is the amount of deformation seen in the basement or subsalt strata
(Figures 9 and 16b). Basement geometries shown in Saura et al. (2014) and Teixell et al. (2017)
are inferred due to lack of exposure. The geometry of the basement graben system beneath the
Amezraï minibasin shown in Saura et al. (2014) was actually modified by Moragas et al. (2017)
based on the results of their physical modeling study. In these natural examples the basement is
shown as flat-topped as sub-salt shortening was taken up by simple fault reactivation and vertical
uplift of hangingwall blocks (Figure 16a, c). If our physical models are indicative of the
deformation intensity one would expect to see in the subsalt basement, then these more pervasive
damage zones could have significant implications for fluid flow and for structural topography at
the base of salt. However, our model basement consisted of essentially cohesionless materials,



and likely does not accurately represent the strength of basement rocks in the High Atlas or
crystalline basement in general. An alternative explanation is that the amount of shortening in
Model 1 was simply far more than that experienced in the Central High Atlas. More work is
required on this topic.


**5.        Concluding Remarks**

Our physical models successfully generated segmented rift systems in a deformable basement
that were subsequently infilled with a salt analog and subjected to further extension and finally
inversion. During extension and subsequent downbuilding diapir and ridge networks formed that
exerted a strong control on deformation styles and patterns during subsequent inversion. Diapir
networks formed primarily in the footwalls of the basement fault system, similar to that
described by Dooley et al. (2005) and Moragas et al. (2017). Diapiric growth was encouraged by
salt expulsion from beneath the subsiding extensional minibasin systems that formed above the
original basement graben, with major diapirs forming consistently in the locations of major relay
systems or interlocking transfer zones that originally separated the basement graben systems.
These more gently dipping structures facilitated more efficient salt expulsion driving diapiric
growth at these locations. Extensional deformation in suprasalt strata was strongly decoupled.

Inversion these salt-bearing rifts produced strongly decoupled shortening belts in basement and
suprasalt sequences. In the suprasalt section deformation geometries and locations were
primarily controlled by the salt diapir network produced doing extension and subsequent



downbuilding with thrusts formed minibasin margins where the overburden was thinnest and
weakest. Extensional minibasins display little or no internal deformation as deformation was
soaked up by diapirs and by these marginal thrusts, in a similar fashion to that observed from teh
Central High Atlas of Morocco. Complex structures form where salt-cored box folds weld shut
by hinge and limb failure. In the subsalt section the structural style is very different consisting of
strongly inverted and pervasively deformed graben systems along with the formation of new
popup structures as these inverted graben locked up. Inversion of these graben uplifted and
welded the composite extensional minibasin system forming an almost flat-topped plateau across
the center of the model. Significant reactivation of graben-bounding faults during inversion was
aided by interstitial infiltration of our salt analog that helped "lubricate" the precursor faults.



**6.       Acknowledgements**

This study was funded by the Applied Geodynamics Laboratory consortium consisting of the
following companies: Anadarko, BHP Billiton, BP, Chevron, Ecopetrol, EMGS, Eni,
ExxonMobil, Fieldwood, Hess, ION Geophysical, Midland Valley, Murphy, Noble Energy,
NOOC, Petrobras, Petronas, PGS, Repsol, Rockfield, Talos, Shell, Spectrum Geo, Stone, Talos,
TGS, Total, WesternGeco, and Woodside. Additional funding for the authors came from the
Jackson School of Geosciences. TD thanks James Donnelly, Nathan Ivivic, Brandon Williamson
and Rudy Lucero for logistical support in the modeling laboratories. Nancy Cottington is thanked
for colorizing the salt in each dip section from Model 1. The authors thank Jaume Vergés (CSIC,
Barcelona) and Grégoire Messager (Equinor, Norway) for initially introducing them to the salt
tectonics of the Moroccan High Atlas. Publication was authorized by the director of the Bureau
of Economic Geology, Jackson School of Geosciences, The University of Texas at Austin.

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



**Figure Captions**

Figure 1. Summary geological map of the central High Atlas of Morocco. Jurassic intrusive
massifs containing upper Triassic shale, basalt and evaporite inliers have been interpreted as
former diapiric ridge that separated extensional minibasins formed during Permian to Early
Jurassic punctuated rifting. AmMB, Amezraï minibasin. AzMB, Azag minibasin. Map redrawn
and modified from Teixell et al., 2017.

Figure 2. Summary of experimental  setup used in models shown in this study. (a) Cross section
view of the pre-rift setup. Models consist of a stretching rubber sheet overlain by a thin basal
detachment and polymer 'slabs' covered by a layered sandpack. (b) Overhead view of
deformation rig prior to emplacement of the layer pre-rift overburden. See text for further details.

Figure 3. (a) Height-change map of Model 1 after pre-salt extension. Three en-échelon graben in
model center are separated by accommodation zones with relays. Marginal graben formed at the
model periphery. (b) Strain map of Model 1 during pre-salt extension. Accommodation zones
consist of interlocking extensional faults. Note that some relays are breached. See text for further
details.

Figure 4. Emplacement of syn-rift salt in Model 1. (a) Pre-salt graben are infilled with our salt
analog. (b) A regional salt fringe is then emplaced across the entire model.





Figure 5. (a) Height-change map during post-salt extension in Model 1. Post-salt extension was
now 4 cm. Note the composite minibasin extending across the model center, above the original
graben system. (b) Strain map of the same increment of post-salt extension. Note the diffuse
strains in the suprasalt cover. Most extensional strains mark outer-arc extension above reactive
diapirs. Note the minor shortening strains within the minibasin due to inner-arc contraction
within the subsiding minibasin.

Figure 6. Height-change maps of Model 1 after 16 (a) and 48 (b) hours of post-extension loading.
In (a) we see the major diapir networks, formed during extension, continuing to rise as salt is
expelled from beneath adjacent minibasins. After 48h loading (b) activity is now focused on two
major diapirs. See main text for more details.

Figure 7. Height-change maps (a-d) reveal the evolution of Model 1 during inversion. Initial
shortening and uplift was focused on the diapirs formed during extension and loading (a),
followed by uplift of the composite minibasin above the model center and rejuvenation of the
diapir and ridge networks (b-d).

Figure 8. Overhead view of Model 1 after 25 cm shortening. Diapirs 1 and 2 are clearly visible in
this view as emergent salt sheets. Section lines are those shown in Figure 9. See text for further
details.

Figure 9. Representative sections through Model 1. Locations are shown on Figure 8. Inset
shows the model stratigraphy.




Figure 10. Details from Model 2 (see Table 1). (a) Height-change map that evolved during Stage
1 of Model 2. (b) Height-change map of Model 2 during post-salt extension. (c) Height-change
map of Model 2 during post-extension loading. (d) Cross section from Model 2 illustrating
extensional minibasins and diapirs of varying heights formed in the footwalls of the subsalt
graben. See text for further details.

Figure 11. Details of Model 3. (a-b) Depth slices through Model 3. (c-e) Arbitrary lines through
a portion of Model 3. (f) Original location of yellow marker 'salt' in Model 3. See text for further
details.

Figure 12. 3D reconstruction of the salt volume from Model 1. See main text for details.

Figure 13. (a) Height-change map from Stage 3 of Model 1 illustrating the diapir and ridge
networks that formed during extension rising during post-extension loading. (b-c) Depth slice
and dip section through Model 1 illustrating the five (a-e) main diapir networks.

Figure 14. Detailed views (a-c) of sections through Model 1 illustrating the evolution of an "S"
structure.

Figure 15. (a) Detailed view of a non-inverted subsalt graben from Model 2. Note the
asymmetric geometry and the formation of a keystone structure. (b) Detailed view of an inverted





subsalt graben from Model 1. Inversion of this graben uplifted and welded the overlying
extensional minibasin.

Figure 16. (a) Cross section through the Amezraï minibasin, Moroccan High Atlas. Note the
uptilted minibasin margins, lack of internal deformation within the minibasin and the complex
flanking diapirs and thrust welds. Redrawn from Moragas et al. (2017). (b) Detailed views of
minibasin margins and associated thrust welds from Model 1. (c) E-W cross section through the
Azag minibasin. Note the thrust welds and tilted nature of the minibasin. Redrawn from Teixell
et al. (2017).



Table 1. Model names and values for extension and inversion for experiments described in the
main text. *denotes basal detachment was limited to the central region of the model.





| Model Name | Basal detachment thickness (cm) | Number of basal slabs and their height (cm) | Regional salt fringe thickness (cm) | Pre-salt Extension (cm) | Post-salt extension (cm) | Total extension (cm) | Inversion (cm) |
|---|---|---|---|---|---|---|---|
| | | | | Stage 1 | Stages 2-3 | | Stage 4 |
| Model 1 | 0.4 | 3 x 1.5 | 1.2 | 3 | 6 | 9 | 25 |
| Model 2 | 0.4* | 3 x 1.5 | 1.2 | 3 | 6 | 9 | 0 |
| Model 3 | 0.4* | 3 x 1.5 | 1.2 | 2.5 | 6.5 | 9 | 8 |

Table 1. Model names and values for extension and inversion for experiments described in the main text. *denotes basal detachment was limited to the central region of the model.





Figure 1







Figure 2





Figure 3



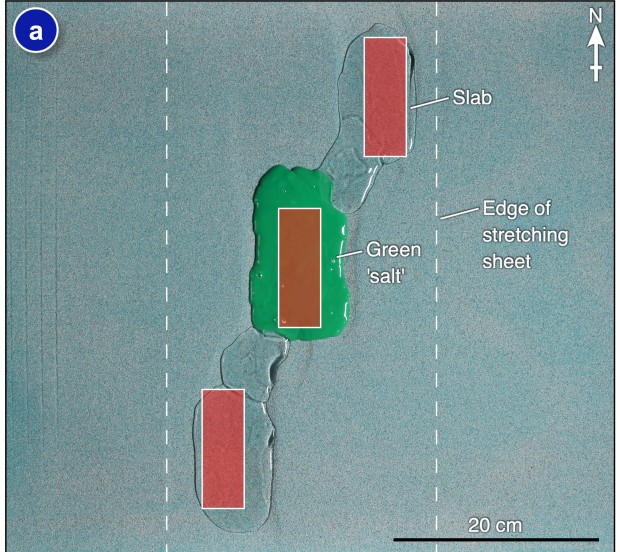
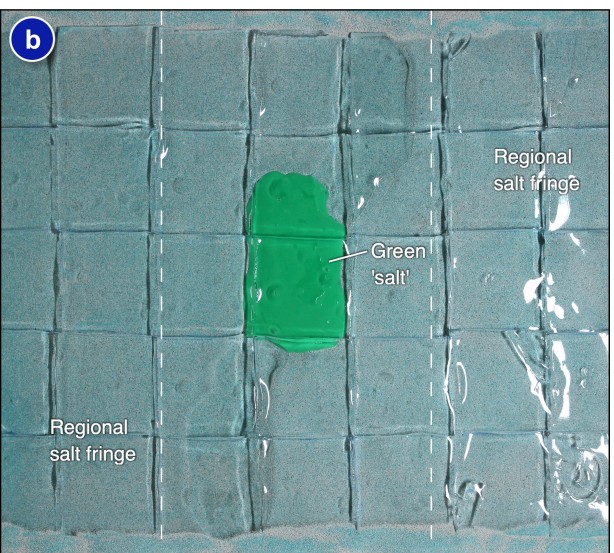

Figure 4





Figure 5



Figure 6



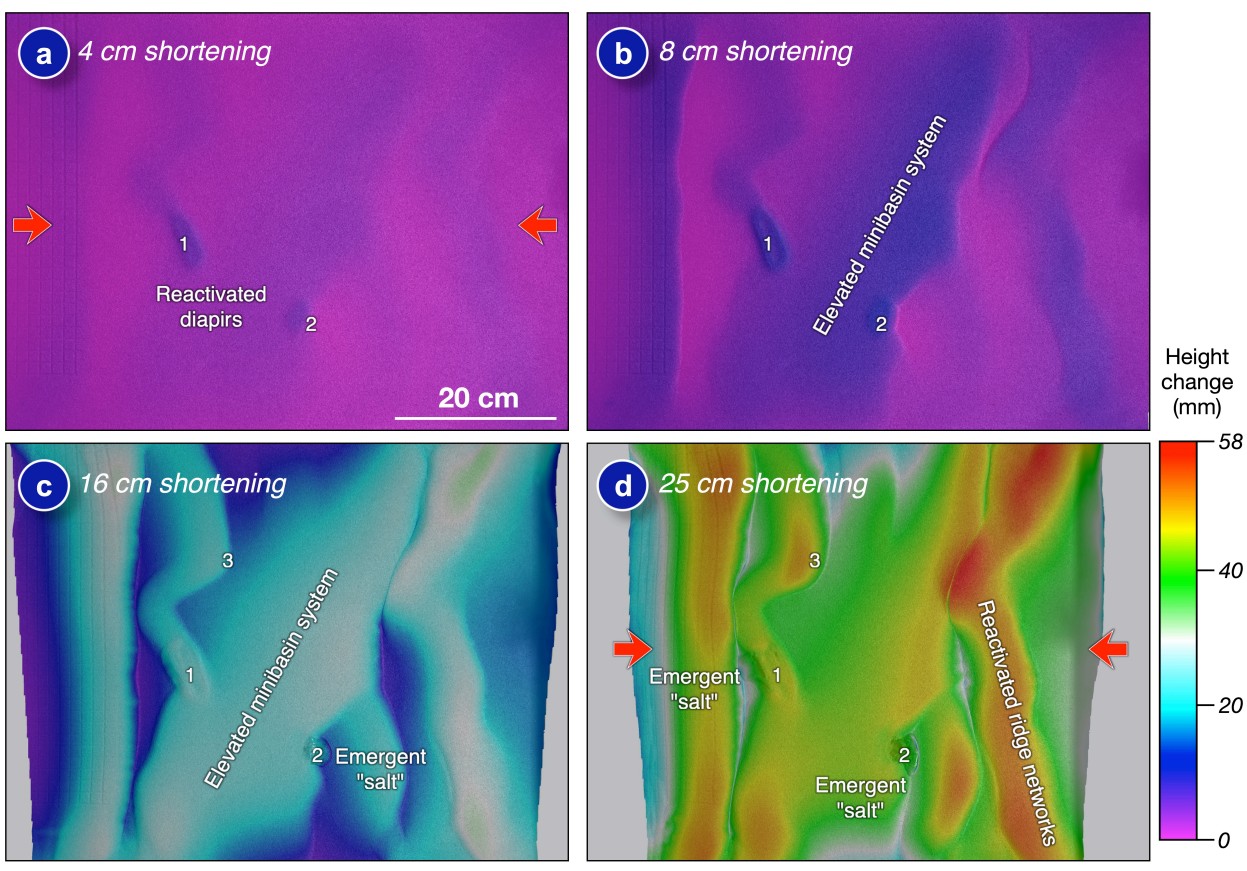

Figure 7





Figure 8







Figure 9



Solid Earth Open Access
Discussions



Figure 10

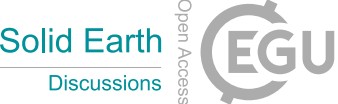



Figure 11





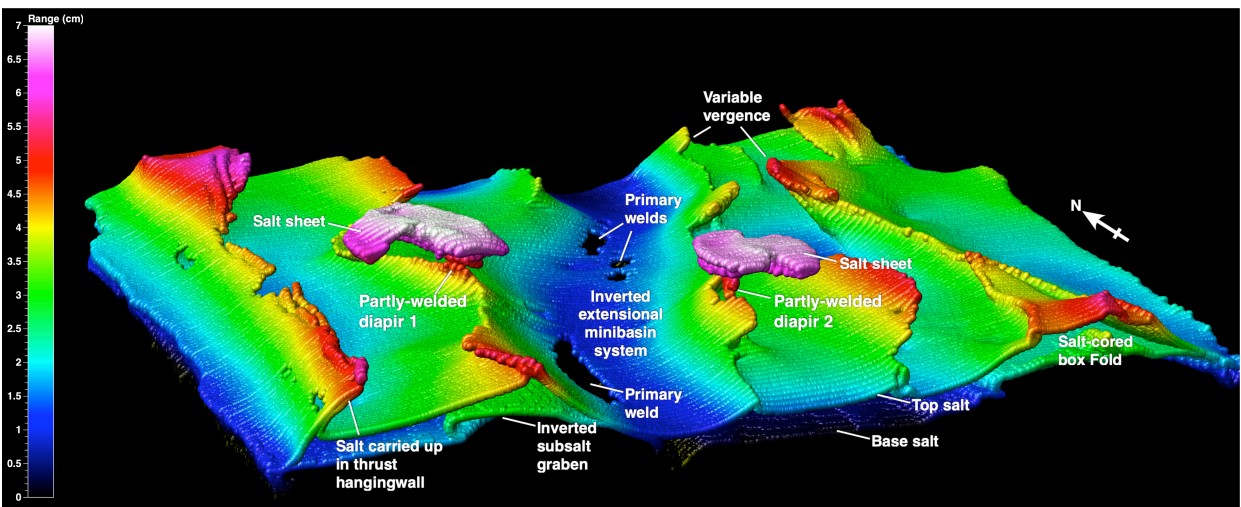

Figure 12



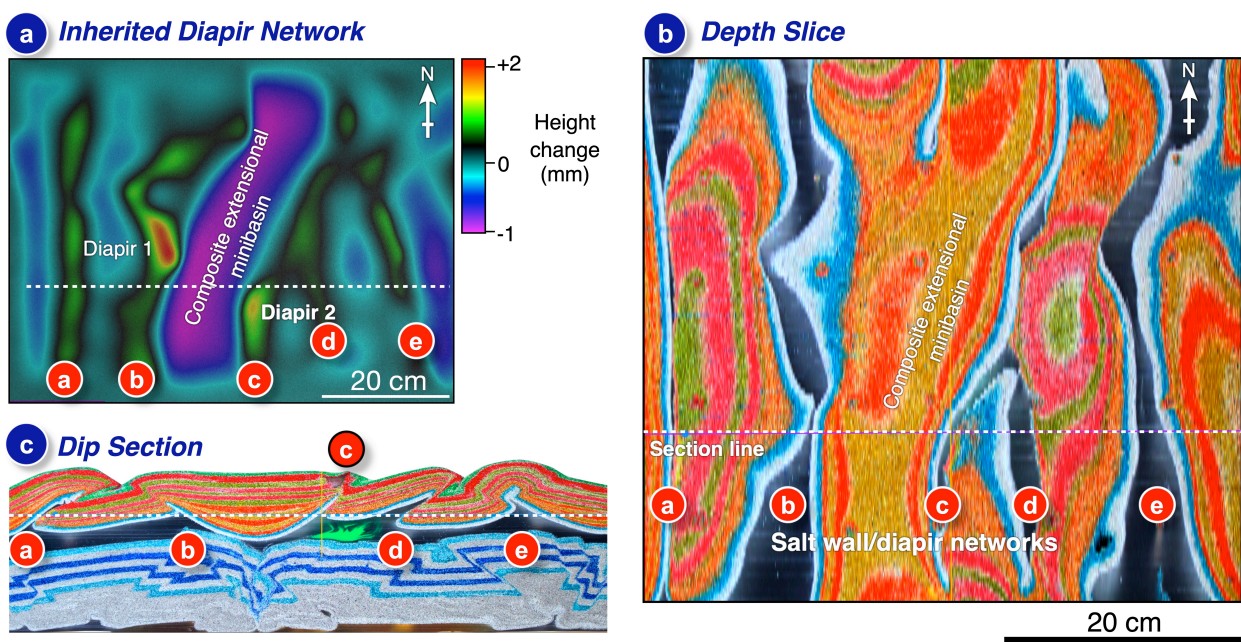

Figure 13



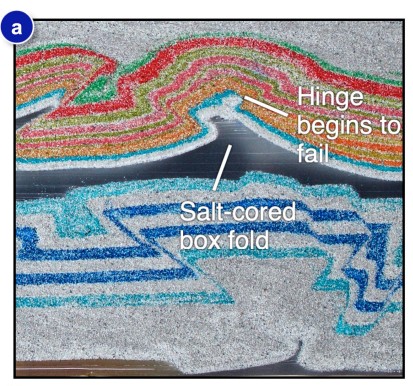

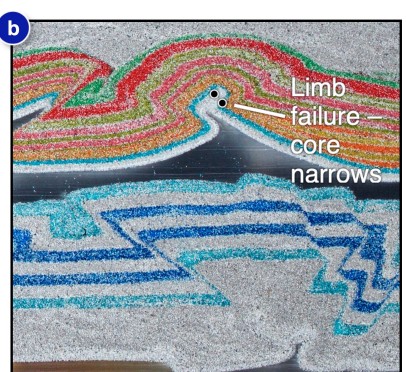

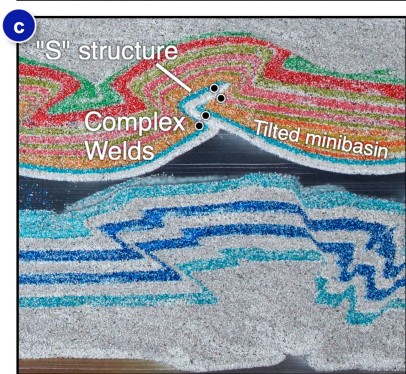

10 cm

Figure 14



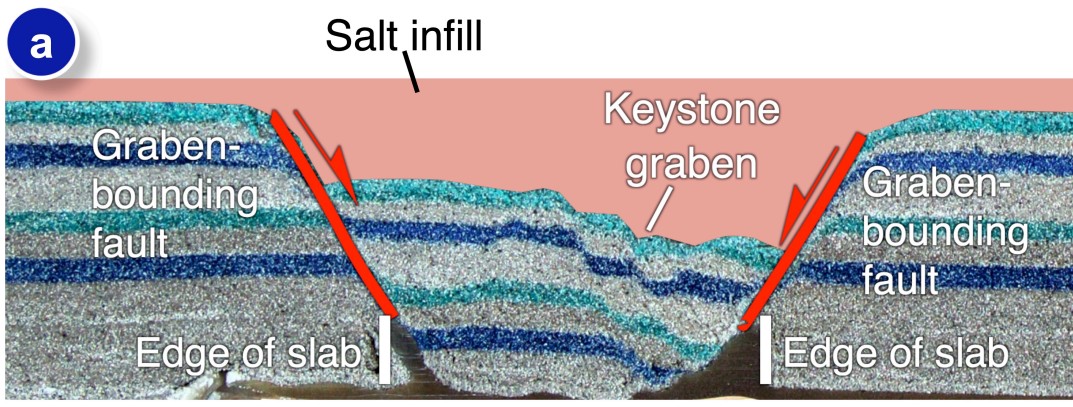

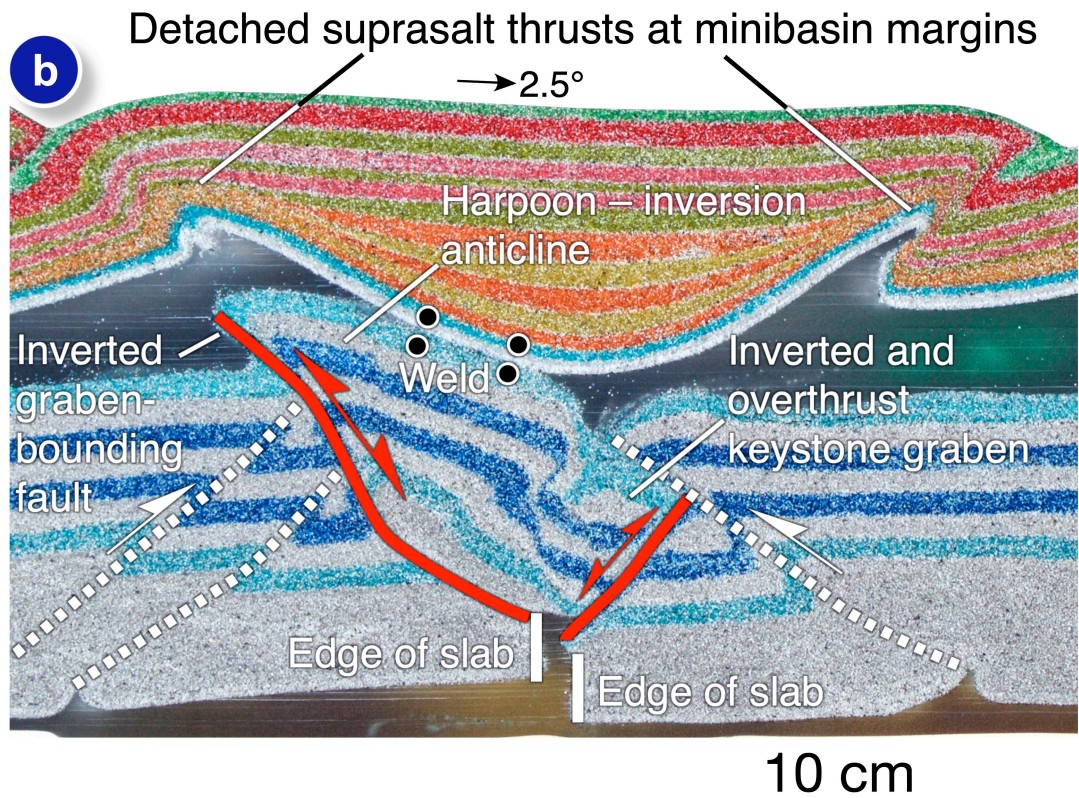

Figure 15





Figure 16