# Peer review of "Extension and Inversion of Salt-Bearing Rift Systems 2 3 Tim P. Dooley & Michael R. Hudec 4 5 Applied Geodynamics Laboratory, Bureau of Economic Geology, Jackson School of 6 Geosciences, University Station Box X, Austin Texas 78713, USA 7 8 Abstract: 9 10 We used physical models to investi"

_Solid Earth, 2020_

## Referee Comment (RC1) · Antonio Teixell (Referee) · 4 Mar 2020

This manuscript uses analog models to investigate the tectonic inversion of salt-bearing extensional grabens. While models of tectonic inversion exist in the literature, some of them including polymer (=salt) layers, the ms. by Dooley and Hudec has the novelty of incorporating a subsalt deformable section, which aims to provide analogues for the compressional structure of pre-salt basements, often a poorly resolved problem in compressed rift systems. In comparison to other published models of compressional systems with multiple "salt" décollement levels (e.g. Couzens et al. 2003 –a paper that should be referenced), Dooley and Hudec's models incorporate early episodes of extensional deformation, which feature synkinematic sedimentation to produce salt migration and diapiric structures (later submitted to compression). No surprise that in the

models there is a marked decoupling between the subsalt and suprasalt units, which puts a warning on the subsurface interpretation of little exhumed, natural inverted rift systems, as the High Atlas of Morocco that is taken as a reference field case. After the early cartoons by Letouzey et al. in 1995, for moderately inverted salt-bearing rifts with no seismic information very often we are tempted to keep in place the parent normal faults (even if reactivated) that we infer as early triggers of observed salt diapirs. The analog models by Dooley and Hudec are welcome in that they remind us that as shortening increases, the connection between the diapirs and the parent faults is likely to be lost. This is likely to be the case in the High Atlas (although with the available data so far it is hard to tell), but also must happen in other basement-involved, salt-detached thrust systems as the northern Pyrenees, where we argued for largely decoupled and displaced salt walls in contrast to autochthonous diapir models (e.g. Labaume and Teixell, Tectonophysics, accepted). The inversion models by Dooley and Hudec provide inspiring images for such natural examples, if the model sand is accepted as a valid analog for crystalline or (non-horizontal) slate basements. Further challenges to the application to natural cases may come, as the authors explicitly recognize, from the tricky simulation of fault-inversion by faulted sand, which most commonly fails to reproduce fault weakening and reactivation. The ms. reads well and is appropriately illustrated, and deserves publication in Solid Earth with minor revision. When analyzing the cross-sectional views of model 1, I had some trouble understanding the inversion features of the subsalt pile, as in Fig. 9 the inverted graben is not so evident (I mean, I did not get an appreciation of how subsalt faults were inverted). Not until I saw Fig. 10 that I got a clear idea (Fig. 14 also helped). The authors may want to consider presenting the uncompressed profiles before actually showing the compressed ones, which in fact represent one step further of an evolution. I also wondered what would happen if there was no salt fringe out of the modeled rift, as actually happens in many natural cases. Fringes cause the post-salt extension to be more diffuse that the first-phase graben system. What happens in basement in this case? On the other hand, Fig. 12 is a very nice polymer (=salt) volume illustration, which shows similarities to

those obtained from natural salt cases after 3D seismic data. Note that minibasins are not always flanked by outward-vergent thrusts as written in line 367 (Fig. 9), which is interesting. Another interesting result is that after shortening, fault footwalls remain broadly inflated (beyond local diapirs). If applicable to nature, this suggests that, counterintuitively, some minibasins may be actually underlain by highest subsalt relief. The application of the model results for the High Atlas cases is preliminary; certainly more analogies can be explored by further work. A natural continuation of the models presented could be including intervening horsts without salt between salt-bearing grabens. I believe that this happens in parts of the High Atlas, such as the Mouguer massif – would that impede major decoupling and translation? The Azag minibasin as drawn looks indeed tilted in a post-depositional stage (although the analog models do not get that much rotation), but note that cases like that are lagged by the absence of subsurface data: there is little control about the stratal geometry at depth and one tends to complete sections in a conservative way. Again, analog models may help in showing the viability of geometric interpretations that may be adopted.

—————————————————————

---

## Referee Comment (RC2) · Oriol Ferrer (Referee) · 16 Mar 2020

Using an experimental approach based on physical models, this manuscript analyses the role of syn-rift evaporites during extension and subsequent inversion of salt-bearing segmented rift basins. Different authors have addressed this type of studies using rigid basement blocs to simulate the basement with the consequent mechanical limitations that this methodology entails. The present manuscript has partly solved this limitation in an original way: a hybrid system that combines a rubber sheet (classically used to constraint the location of extensional faults during stretching) with polymer slabs (also used in physical modelling as "seeds" to constraint fault nucleation). This hybrid experimental setup allows to achieve a significant degree of inversion of pre-salt grabens (an inherent issue of the models that use rigid blocks). In addition, the en-echelon

distribution of the different slabs allows to simulate a segmented rift system. Another interesting point addressed in the manuscript is how syn-rift salt controls the structural style during extension and consequently, how the inherited salt structures at the end of the extension constraints deformation during inversion.

The manuscript reads well and the quality of figures is excellent, they perfectly illustrate the text. The scaling of the experimental program is correct and the analog materials are the classical ones used in physical modelling of salt tectonics. The experimental results are compared with natural examples from the Moroccan High Atlas but they would be perfectly applicable to other fold-and-thrust belts involving inverted salt-bearing rift basins such as the Pyrenees. I am sure this manuscript will be useful to the understanding of inverted salt-bearing rift basins. For this reason, I recommend its publication in Solid Earth journal after few minor revisions (please, see below general and specific comments, suggestions and questions).

Best regards,

Oriol Ferrer

General comments

A point that I consider should be implemented in section "2.1. Model Design and Scaling" is as far as syntectonic sedimentation is concerned. What is the sedimentation rate? Did you keep the pre-extensional regional fixed? How much did you raise it every new synkinematic layer of sand? These are points that the reader should know. These points are addressed in the experimental results section, but they should be moved at section 2.1.

Include a figure like figure 2 of Roma et al. (2018b) could help to understand the procedure applied during model run. This is just a suggestion.

As far as setup is concerned, I don't understand why did you modify the extension of the basal detachment layer in models 1 and 2/3 respectively. Why not to use the same

for the 3 experiments? Can this modification influence the final results in any way?

The section 3 (Experimental results) is clearly described and well ordered. However, I disagree about the harpoon structure described in lines 28-287 (also in line 419, section 4.3. "Shortening in the subsalt section"). According to McClay (1995), the inversion of the wedge shaped synextensional strata produces typical "harpoon" or "arrowhead" geometries, the shape of which depends upon the geometry of the underlying extensional faults. In these lines of the manuscript, this geometry is wrongly applied to the inverted preextensional unit. Please, modify it.

Regarding this section, could the marginal grabens be related to edge effects of your experimental setup?

As it has been pointed out in the manuscript, the contractional reactivation of graben-bounding faults during inversion can be favored by the polymer infiltration into the granular material (sand). This infiltration occurs either at the interphases between polymer and pre- / supra-polymer sand during the setup of the experiments, and throughout the experiment when new surfaces (faults) developed. This process occurs when the sand-polymer interphase is preserved for a long time, so can we interpret that the longer the contact, the wider the area affected by polymer infiltration? If this is the case, the contractional reactivation of the sand-polymer interphase in the fault will be more effective because it will "lubricate" in a more efficient way. Is this observation true? Other prompt questions regarding this topic are: What is the infiltration rate? There is any control of this process during the construction/run of the experiments? What are the main factors that control it? I perfectly understand that these questions are out of the scope of the manuscript, but as a modeler and after to notice similar processes, I consider that this topic should be discussed in the manuscript.

Considering that model 2 was sliced at the end of the extensional stage, and model 1 is similar but with 25 cm of inversion, it would be interesting to include a section discussing the contractional reactivation of primary welds and what is the role that they

play during inversion. Are they reactivated as thrust welds? What occurs with their surface? There is any increasing/decreasing on their surface during inversion? Have you noticed the opening of the primary welds during inversion in your 3D voxels? The paper by Roma et al. (2018b) includes some discussion about this topic that could be compared with the models included in the present manuscript.

Due to the few published works in analog modeling addressing the role of synrift evaporites during extension and subsequent inversion, I consider that some additional references such Soto et al. (2007); Ferrer et al. (2014); Roma et al. (2018b) should be included in the manuscript.

Ferrer et al. (2014). The role of salt layer in the hangingwall deformation of kinked-planar extensional faults: Isights from 3D analogue models and comparison with the Parentis Basin. Tectonophysics, 636, 338-350.

Roma et al. (2018b). Weld kinematics of syn-rift salt during basement-involved extension and subsequent inversion: Results from analog models. Geologica Acta, 16 (4), 391-410.

Soto et al. (2007). Geometry of half-grabens containing a mid-level viscous décollement. Basin Research, 19, 437-450.

Specific comments:

Line 50 "Calloway" should be "Callaway" Line 74 at the end of the line, modify "halo kinetic" for "halokinetic" Lines 109-110 Check the sentence Line 113 indicate the thickness of the basal polymer Line 114 indicate the dimensions of the slabs Line 157 indicate the % of sand and ceramic microspheres that you used, and if this % is in weight or volume Line 508 modify "teh" for "the" at the end of the line

Figure 2 is difficult to understand. Is the rubber sheet transparent?

Figure 4 Why did you use green polymer? I did not found the explanation in the manuscript.

The word "stage" is used indistinctly in capital or lowercase to refer to the different evolutionary stages of the models. Please, use uniform criteria.

There are some references that do not match those of the reference list. Among those I have detected: Bonini et al. (2011); Moragas et al. (2016); and Martín-Martín et al. (2016). Please, check which one is correct and unify. Similarly, there are missing references in the reference list: Adam et al., 2005 (line 180); Sibson, 1995 (line 430) and Anderson, 1951 (line 431).

---

## Author Response (AR1)

Dear Piotr,

Please find our responses to reviewer comments and the revised paper and figures of our manuscript "Extension and Inversion of Salt-Bearing Rift Systems", for inclusion in the special volume "Inversion tectonics – 30 years later" in Solid Earth.

Let me first address the reviewers' comments.

Comments by Antonio Teixell:

This manuscript uses analog models to investigate the tectonic inversion of salt-bearing extensional grabens. While models of tectonic inversion exist in the literature, some of them including polymer (=salt) layers, the ms. by Dooley and Hudec has the novelty of incorporating a subsalt deformable section, which aims to provide analogues for the compressional structure of pre-salt basements, often a poorly resolved problem in compressed rift systems. In comparison to other published models of compressional systems with multiple "salt" décollement levels (e.g. Couzens et al. 2003 –a paper that should be referenced), Dooley and Hudec's models incorporate early episodes of extensional deformation, which feature synkinematic sedimentation to produce salt migration and diapiric structures (later submitted to compression). No surprise that in the models there is a marked decoupling between the subsalt and suprasalt units, which puts a warning on the subsurface interpretation of little exhumed, natural inverted rift systems, as the High Atlas of Morocco that is taken as a reference field case. After the early cartoons by Letouzey et al. in 1995, for moderately inverted salt-bearing rifts with no seismic information very often we are tempted to keep in place the parent normal faults (even if reactivated) that we infer as early triggers of observed salt diapirs. The analog models by Dooley and Hudec are welcome in that they remind us that as shortening increases, the connection between the diapirs and the parent faults is likely to be lost. This is likely to be the case in the High Atlas (although with the available data so far it is hard to tell), but also must happen in other basement-involved, salt-detached thrust systems as the northern Pyrenees, where we argued for largely decoupled and displaced salt walls in contrast to autochthonous diapir models (e.g. Labaume and Teixell, Tectonophysics, accepted). The inversion models by Dooley and Hudec provide inspiring images for such natural examples, if the model sand is accepted as a valid analog for crystalline or (non-horizontal) slate basements. Further challenges to the application to natural cases may come, as the authors explicitly recognize, from the tricky simulation of fault-inversion by faulted sand, which most commonly fails to reproduce fault weakening and reactivation. The ms. reads well and is appropriately illustrated, and deserves publication in Solid Earth with minor revision.

When analyzing the cross-sectional views of model 1, I had some trouble understanding the inversion features of the subsalt pile, as in Fig. 9 the inverted graben is not so evident (I mean, I did not get an appreciation of how subsalt faults were inverted). Not until I saw Fig. 10 that I got a clear idea (Fig. 14 also helped). The authors may want to consider presenting the uncompressed profiles before actually showing the compressed ones, which in fact represent one step further of an evolution. I also wondered what would happen if there was no salt fringe out of the modeled rift, as actually happens in many natural cases. Fringes cause the post-salt extension to be more diffuse that the first- phase graben system. What happens in basement in this case? On the other hand, Fig. 12 is a very nice polymer (=salt) volume illustration, which shows similarities to those obtained from natural salt cases after 3D seismic data. Note that minibasins are not always flanked by outward-vergent thrusts as written in line 367 (Fig. 9), which is interesting. Another interesting result is that after shortening, fault footwalls remain broadly inflated (beyond local diapirs). If applicable to nature, this suggests that, coun- terintuitively, some minibasins may be actually underlain by highest subsalt relief. The application of the model results for the High Atlas cases is preliminary; certainly more analogies can be explored by further work. A natural continuation of the models pre- sented could be including intervening horsts without salt between salt-bearing grabens. I believe that this happens in parts of the High Atlas, such as the Mouguer massif – would that impede major decoupling and translation? The Azag minibasin as drawn looks indeed tilted in a post-depositional stage (although the analog models do not get that much rotation), but note that cases like that are lagged by the absence of subsur- face data: there is little control about the stratal geometry at depth and one tends to complete sections in a conservative way. Again, analog models may help in showing the viability of geometric interpretations that may be adopted.

Our reply to Antonio:

Many thanks for the positive review of our inversion paper. You had some specific and general comments that I will address below.

"In comparison to other published models of compressional systems with multiple "salt" décollement levels (e.g. Couzens et al. 2003 –a paper that should be referenced), Dooley and Hudec's models incorporate early episodes of extensional deformation, which feature synkinematic sedimentation to produce salt migration and diapiric structures (later submitted to compression)."

In reality in these models the lower decollement is simply that, a decollement to ensure that shortening is transferred across the rift system. In models where by the thin lower decollement was not present across the entire system the result was a shortcut fault transferring minor shortening up to the outer edges of the suprasalt sequence (Model 3). But I agree that Couzens et al. should be referenced, along with selected other papers with multiple detachment levels, for completeness.

"The inversion models by Dooley and Hudec provide inspiring images for such natural examples, if the model sand is accepted as a valid analog for crystalline or (non- horizontal) slate basements. Further challenges to the application to natural cases may come, as the authors explicitly recognize, from the tricky simulation of fault-inversion by faulted sand, which most commonly fails to reproduce fault weakening and reactivation."

Yes, I tried to temper the arguments in this manuscript as our model materials (sands) in the subsalt section may not reflect the "strength" of basement rocks in these orogens. However, we believe that, and as noted by you, that these models provide examples of possible basement deformation scenarios in areas where there is generally little or no basement exposure nor seismic data to aid interpretation.

"The authors may want to consider presenting the uncompressed profiles before actually showing the compressed ones, which in fact represent one step further of an evolution. I also wondered what would happen if there was no salt fringe out of the modeled rift, as actually happens in many natural cases. Fringes cause the post-salt extension to be more diffuse that the first- phase graben system. What happens in basement in this case?"

I pondered using a different order of presenting the 3 models when initially writing the manuscript. But I found the text flowed better when Model 1 could be described fully before delving into the details of the deformation (both pre- and post-inversion) with the use of Models 2 and 3 in a more discussion-like section. Your point on the salt fringe is well taken. Yes, this is likely to results in highly variable deformation styles across the inverted rift system – a mixture of coupled and decoupled geometries further adding to the complexity. We have done some work on this but more needs to be done, which could be applied to other areas in the High Atlas as you mention toward the end of your comments. A sentence or two will be added on this topic in the revised manuscript. One thing we noted is that without a significant salt fringe it was difficult to produce diapirs on the flanks of the segmented graben systems.

"Note that minibasins are not always flanked by outward-vergent thrusts as written in line 367 (Fig. 9), which is interesting. Another interesting result is that after shortening, fault footwalls remain broadly inflated (beyond local diapirs). If applicable to nature, this suggests that, counterintuitively, some minibasins may be actually underlain by highest subsalt relief."

For the most part minibasins are flanked by outward-vergent thrusts but you correct there are a few locations along the main rift system that are not. The text will be revised accordingly. Yes, the highest subsalt structural topography lies below the minibasins which is fascinating. I think the height-change maps showing the relief development during inversion quite spectacularly illustrate this with the minibasin system being ele- vated by this subsalt inversion, and with quite a low degree of rotation of the minibasin strata – more on that below.

"The Azag minibasin as drawn looks indeed tilted in a post-depositional stage (although the analog models do not get that much rotation), but note that cases like that are lagged by the absence of subsurface data: there is little control about the stratal ge- ometry at depth and one tends to complete sections in a conservative way. Again, analog models may help in showing the viability of geometric interpretations that may be adopted."

Yes, the models give possible answers to subsurface geometries and the processes that went into making them the way they are. But, they are just that, models. But there are sections from Model 1 that do illustrate significant rotation and Figure 16 will be altered to include an example of this. The original talk I gave on this model series had such a model example and you reminded me of that. Thanks!

Comments by Oriol Ferrer:

Using an experimental approach based on physical models, this manuscript analyses the role of syn-rift evaporites during extension and subsequent inversion of salt-bearing segmented rift basins. Different authors have addressed this type of studies using rigid basement blocs to simulate the basement with the consequent mechanical limitations that this methodology entails.
The present manuscript has partly solved this limitation in an original way: a hybrid system that
combines a rubber sheet (classically used to constraint the location of extensional faults during
stretching) with polymer slabs (also used in physical modelling as "seeds" to constraint fault
nucleation). This hybrid experimental setup allows to achieve a significant degree of inversion of
pre-salt grabens (an inherent issue of the models that use rigid blocks). In addition, the en-
echelon distribution of the different slabs allows to simulate a segmented rift system. Another
interesting point addressed in the manuscript is how syn-rift salt controls the structural style
during extension and consequently, how the inherited salt structures at the end of the extension
constraints deformation during inversion.
The manuscript reads well and the quality of figures is excellent, they perfectly illustrate the text.
The scaling of the experimental program is correct and the analog materials are the classical ones
used in physical modelling of salt tectonics. The experimental re- sults are compared with natural
examples from the Moroccan High Atlas but they would be perfectly applicable to other fold-
and-thrust belts involving inverted salt-bearing rift basins such as the Pyrenees. I am sure this
manuscript will be useful to the under- standing of inverted salt-bearing rift basins. For this
reason, I recommend its publi- cation in Solid Earth journal after few minor revisions (please,
see below general and specific comments, suggestions and questions).
Best regards,
Oriol Ferrer
General comments:
A point that I consider should be implemented in section "2.1. Model Design and Scaling" is as
far as syntectonic sedimentation is concerned. What is the sedimentation rate? Did you keep the
pre-extensional regional fixed? How much did you raise it every new synkinematic layer of
sand? These are points that the reader should know. These points are addressed in the
experimental results section, but they should be moved at section 2.1.
Include a figure like figure 2 of Roma et al. (2018b) could help to understand the procedure
applied during model run. This is just a suggestion.
As far as setup is concerned, I don't understand why did you modify the extension of the basal
detachment layer in models 1 and 2/3 respectively. Why not to use the same for the 3
experiments? Can this modification influence the final results in any way?
The section 3 (Experimental results) is clearly described and well ordered. However, I disagree
about the harpoon structure described in lines 28-287 (also in line 419, sec- tion 4.3. "Shortening
in the subsalt section"). According to McClay (1995), the inversion of the wedge shaped
synextensional strata produces typical "harpoon" or "arrowhead" geometries, the shape of which
depends upon the geometry of the underlying extensional faults. In these lines of the manuscript,
this geometry is wrongly applied to the inverted preextensional unit. Please, modify it.
Regarding this section, could the marginal grabens be related to edge effects of your
experimental setup?

As it has been pointed out in the manuscript, the contractional reactivation of graben- bounding
faults during inversion can be favored by the polymer infiltration into the granular material
(sand). This infiltration occurs either at the interphases between polymer and pre- / supra-
polymer sand during the setup of the experiments, and throughout the experiment when new
surfaces (faults) developed. This process occurs when the sand- polymer interphase is preserved
for a long time, so can we interpret that the longer the contact, the wider the area affected by
polymer infiltration? If this is the case, the con- tractional reactivation of the sand-polymer
interphase in the fault will be more effective because it will "lubricate" in a more efficient way.
Is this observation true? Other prompt questions regarding this topic are: What is the infiltration
rate? There is any control of this process during the construction/run of the experiments? What
are the main factors that control it? I perfectly understand that these questions are out of the
scope of the manuscript, but as a modeler and after to notice similar processes, I consider that
this topic should be discussed in the manuscript.
Considering that model 2 was sliced at the end of the extensional stage, and model 1 is similar
but with 25 cm of inversion, it would be interesting to include a section discussing the
contractional reactivation of primary welds and what is the role that they play during inversion.
Are they reactivated as thrust welds? What occurs with their surface? There is any
increasing/decreasing on their surface during inversion? Have you noticed the opening of the
primary welds during inversion in your 3D voxels? The paper by Roma et al. (2018b) includes
some discussion about this topic that could be compared with the models included in the present
manuscript.
Due to the few published works in analog modeling addressing the role of synrift evaporites
during extension and subsequent inversion, I consider that some additional references such Soto
et al. (2007); Ferrer et al. (2014); Roma et al. (2018b) should be included in the manuscript.
Ferrer et al. (2014). The role of salt layer in the hangingwall deformation of kinked- planar
extensional faults: Isights from 3D analogue models and comparison with the Parentis Basin.
Tectonophysics, 636, 338-350.
Roma et al. (2018b). Weld kinematics of syn-rift salt during basement-involved exten- sion and
subsequent inversion: Results from analog models. Geologica Acta, 16 (4), 391-410.
Soto et al. (2007). Geometry of half-grabens containing a mid-level viscous décolle- ment. Basin
Research, 19, 437-450.
Specific comments:
Line 50 "Calloway" should be "Callaway" Line 74 at the end of the line, modify "halo kinetic"
for "halokinetic" Lines 109-110 Check the sentence Line 113 indicate the thick- ness of the basal
polymer Line 114 indicate the dimensions of the slabs Line 157 indicate the % of sand and
ceramic microspheres that you used, and if this % is in weight or volume Line 508 modify "teh"
for "the" at the end of the line
Figure 2 is difficult to understand. Is the rubber sheet transparent?
Figure 4 Why did you use green polymer? I did not found the explanation in the manuscript.

The word "stage" is used indistinctly in capital or lowercase to refer to the different evolutionary
stages of the models. Please, use uniform criteria.
There are some references that do not match those of the reference list. Among those I have
detected: Bonini et al. (2011); Moragas et al. (2016); and Martín-Martín et al. (2016). Please,
check which one is correct and unify. Similarly, there are missing references in the reference list:
Adam et al., 2005 (line 180); Sibson, 1995 (line 430) and Anderson, 1951 (line 431).

Our reply to Oriol:
Oriol,
Many thanks for your thorough and positive review of our inversion paper. I will reply to your
comments below, pasting in your text in order to answer specific comments if needed.
General comments:
1. Agree, will add more specific information on sedimentation rates etc. in the modeling
methodology section. In general the height-change maps generated from the DIC system guided
me here during all stages of the model runtimes. Base-level was raised just enough to clear any
rising diapirs resulting in sequences that markedly thinned across the rift flanks.
2. This would be more difficult to do for a 3D system that is being modeled here. I think the
figures as they stand give the readers a walk through of the model evolution and final
geometries.
3. Ah! Yes. These experiments were not run in the order they are presented in! Models 2 and 3
were run before Model 1, and it became obvious that the thin basal decollement (3-4 mm of
silicone polymer) needed to cover the entire base of the rig to facilitate inversion of the entire
subsalt and supra salt sequence. The only difference this made to the pre-inversion geometries
was pushing the marginal graben systems further towards the periphery of the model. These
marginal graben are quite minor and the main focus is aimed at the main segmented rift system.
4. Yes, the "harpoon" reference is poor and will be removed. "Inverted subsalt graben" will be
used or something to that effect.
5. Marginal graben – see point 3 above. It's minor and not our focus.
6. This is a very interesting and modeling-focused point. In general the infiltration "depth" I
believe is governed by the grain size of the sediments, and thus by the mixture of
sands/censopheres used in these strata. You have likely seen the same phenomena, and I keep
some of this material in my labs to show visitors that sand can fold if it has been "infiltrated" by
polymer, but still fail in a brittle fashionunder higher strain rates! I really can't answer all your
questions in this section. I'm sure you have seen that welds between sub- and suprasalt sand
layers appear to be very clean, unlike may between sand and a rigid, non-porous baseboard. But
you commonly see a slight change in color of the strata just above and below the weld for about
2-3 mm, again depending on grain size. I will try to add in a sentence or two to clarify this, but
for now it's a known unknown, but the best explanation I have for efficient reactivation of these
subsalt graben.

7. This is a great observation and something I will discuss. If you compare Model 1 (Figures 9
and 15b) with Model 2 (Figure 10) the primary welds are in different locations - much more
toward the minibasin centers in inverted models, but at the flanks of the minibasins in non-
inverted models. Yes, these have likely been "sheared off" during inversion as the entire
minibasin is pushed upwards above the rising hanging walls of the segmented graben systems.
And the length or extent of the welds becomes greater – e.g. see Section 33 in Figure 9. I'll add
in Roma et al. *2018b) and discuss in the relevant section. Thanks.
8. Yes, additional references will be added as per your suggestion. Makes it more complete.
Specific comments:
Will fix those and add reference to volumetric percentages of sand and ceonspheres where
appropriate.
Figure 2 – the stretching rubber sheet is black and is visible through the basal thin polymer layer.
But I have it as green in Figure 2a. I'll adjust this and fix the caption to clarify.
Figure 4 – I though I did mention it. I'll check. This is to track where our model salt in the
central graben flows to during extension and loading.
I will standardize on "Stage" to make it consistent.
Thanks for catching the references.
Again, many thanks for the comments and suggestions/discussions.
Paper revisions:
Again, many thanks to both Antonio and Oriol for their positive and comprehensive reviews of
the paper. All of their comments have been addressed above and with revisions to main text and
to some of the Figures. For example, the modeling setup figure was altered so the stretching
rubber sheet seen in the overhead photo of Figure 2b matches the color of its representation in
Figure 2a. In Figure 16 I added a detail of one cross section through Model 1 showing a
tectonically tilted minibasin with a hypothetical erosion level line. I also did some general
tightening of the text where I saw fit.
Also included in the revised text are clarifications to model setup and procedures such as the
addition of synkinematic sediments. Additionally text has been added in the discussion
addressing Antonio's comments on "connectivity" of salt in such basins. Indeed, there is plenty
of work that needs to be done to address this and compare and contrast inverted salt-bearing
basins with and without a salt fringe that, as noted, aids mobility. Oriol's comments on the
"polymer-sand rind" seen in the models was a very modeling-specific query but I have added a
short sentence in that section addressing this. In fact it led to a long discussion between Oriol and
I and he sent images of similar "polymer-sand rinds" in his modeling work. In addition, his
comments on the welds was great, as the position of primary welds between Model 1 and Model
2 are very different, due to primary welds formed in extension being shear off during inversion,
shifting the locations of the welds. This has been addressed in the revised version of the paper.
Most of these revisions were added in the Discussion section rather than in the Conclusions to keep the final words the reader sees as being a summary of the model findings rather than a
litany of problems with the models themselves.
I hope that the revised manuscript proves acceptable for publication in the special volume
"Inversion tectonics – 30 years later".
Sincerely,

Tim Dooley
30th April 2020

[revised manuscript text omitted]